# OpenReview forum: "STEALTH: Secure Transformer for Encrypted Alignment of Latent Text Embeddings via Semantic Isomorphism Enforcement (SIE) Loss Function"
_TMLR — Accepted by TMLR_

### Review · Reviewer_jkzP · 2025-12-10

**Summary Of Contributions:**

This paper contributes a new strategy for secure text processing under an authorized key encryption model. The idea consists of using a transformer under a custom loss function to embed cyphertexts (in the context of the keys used to generate them) into a representation space which preserves ordering of plaintext similarity, and ensures that encryptions of the same plaintext using different keys are tightly clustered. The paper evaluates this method primarily experimentally, with some theoretical consideration in the appendices.

**Audience:**

Yes

**Audience Explanation:**

I have only a passing acquaintance with cryptography and no knowledge as it applies to machine learning. I'm not sure whether the "authorized key" threat model considered here is ecologically valid and no references are provided within the paper. Providing the cypher keys seems like a huge assumption; I suppose this is meant to be used in cases where the keys do not directly allow the plaintext to be decoded. In any case, the fact that a transformer can learn to invert a pseudorandom number generator using a corpus is an interesting finding in its own right.

**Broader Impact Concerns:**

Broader impacts seem mostly positive, although there is no statement to that effect -- and the ethical considerations around facillitating "secure" (in the cryptographic sense) processing of people's sensitive information by LLMs are worth discussing.

**Claims And Evidence:**

No

**Claims Explanation:**

The experimental results which are included in the main body are quite brief. The recovery accuracy is very impressive -- but things should be re-arranged to bring the more in-depth experimental evaluation out of the massive appendix and into the main body.

I mostly found the exposition clear, although there are some serious weak points. In particular, what the actual setup is not very well-explained and I had to piece it together using material from the appendices: Basically, the idea is to decode cyphertext and key pairs by embedding them with the transformer, and then compare against embeddings of a corpus of domain-relevant text to find near neighbors which serve as the decoded plaintext. The text and notation in the sub-sections 3.1 and 3.2 is quite misleading about this, and the decoding procedure is not clearly explained except in the appendix.

**Requested Changes:**

* Critically, clarify the introduction and methodology sections to explain the method more clearly.

* Critically, re-organize the paper to include more experimental evaluation of the method in the main body.

---

> ### Author Response · Authors · 2025-12-31
> **Response to Reviewer jkzP**
>
> We sincerely thank Reviewer jkzP for the insightful feedback. **We first wish to apologize for the delay in this response, which was caused by an unexpected medical emergency.** We value your recognition of our recovery accuracy and have carefully addressed your concerns regarding methodology and ecological validity.
>
> **1. Experimental Results & Organization (Section 6)**
>
> * **Concern:** Main body results are too brief; critical data is buried in the appendix.
> * **Response:** To address this, we added **two comprehensive aggregate figures** to the main **Results (Section 6)**:
> * **Figure 3 (Fidelity):** A bar chart visualizing BLEU-4 scores across **all 44 datasets**.
> * **Figure 4 (Efficiency):** A corresponding analysis of Inference Latency across all domains.
> These figures allow readers to immediately see the model's robustness and the "Fidelity vs. Efficiency" trade-off across the full experimental suite (GLUE, Medical, Legal, Finance, etc.) without needing to flip to the appendix.
>
> **2. Methodology Clarification (Section 3)**
>
> * **Concern:** Setup and decoding procedure were unclear/misleading in the main text.
> * **Response:** We have completely rewritten **Section 3 (Methodology)**:
> * **Roadmap:** Added a paragraph outlining the logical flow (Operational Scope  Architecture  Inference).
> * **Inference Procedure (Section 3.3):** We added a dedicated subsection explicitly defining the "decoding" process as a **Nearest Neighbor Retrieval** task () against the reference index. The exact mathematical formulation is now front-and-center.
> * **Notation:** We standardized notation to clearly distinguish between the encryption function , model mapping , and reference encoder .
>
> **3. Ecological Validity / Authorized Key (Threat Model)**
>
> * **Concern:** Providing keys seems like a huge assumption; ecological validity?
> * **Response:** We expanded the **Introduction** to contextualize this model within **Confidential Computing (TEEs)** and **Enterprise Search**:
> * **Ecological Niche:** In secure medical search, an authorized user (e.g., doctor) provides a transient key to a secure enclave to query an encrypted database. This is the industry standard for high-utility privacy (distinct from the "untrusted adversary" model of FHE).
> * **References:** We added citations to recent work on hardware-assisted isolation (e.g., Azure Confidential Computing, NVIDIA TEEs) to validate this "Authorized Key" assumption as a standard practice for performance-critical privacy tasks.
>
> **4. Broader Impact (Section 6)**
>
> * **Concern:** Ethical considerations need discussion.
> * **Response:** We added a dedicated **Broader Impact Statement (Section 6)** discussing:
> * **Positive:** Unlocking siloed sensitive data (HIPAA/Finance) for safe analysis.
> * **Ethical Risks:** We warn against "over-reliance" on embedding security. While secure under our threat model, users must not mistake encrypted embeddings for information-theoretic security. We emphasize that access controls to the *retrieval corpus* remain critical.
>
> We believe these revisions significantly strengthen the paper's clarity and contribution.

---

### Review · Reviewer_gCkq · 2025-12-10

**Summary Of Contributions:**

This paper introduces STEALTH, a plug-and-play framework, built to address the existing limitations of computational limits, poor generation quality and information leakage. It attempts to use representation alignment to provide safeguards, ensuring efficiency, isomorphism and composibility. The writing is well-founded and understandable. There is also an interesting architecture at use, although a bit computationally complex, for aggregation of keys.

---

### Weaknesses:
- If I'm correct, STEALTH produces a latent vector, which acts as an input to the LLM. Current LLM vendors do not accept embeddings as an input and hence, the only situation when this would be useful is for open-source models, hosted on private infrastructure. If hosted on private infrastructure and these LLMs are open-source, I do not see the major motivation of STEALTH. Can the authors clarify this?
- I am very impressed with the author's results on all the benchmarks, but I'm curious about how inference works. Looking at Algorithm 1, I see that the authors essentially are computing the nearest neighbor from a given dataset. That given dataset contains original embeddings and we are comparing the embedding with the encrypted embedding. For this to work well, the plaintext corpus must already contain a close match to the encrypted input. Otherwise, the nearest neighbor could be semantically wrong or noisy. This significantly weakens the paper since it seems to be recovering from a known corpus (i.e. this is more retrieval, instead of generation).
- I also fail to see how this $f_\theta$ sufficiently correlates to the given embedding function g, apart from one of the loss functions.

---

**Audience:**

Yes

**Audience Explanation:**

Yes, I think this is definitely an interesting line of work that would interest the community of AI Safety. However, I feel like the paper's motivation should be more well-founded and motivated better.

**Claims And Evidence:**

Yes

**Claims Explanation:**

Majority of the claims in the paper have been experimentally verified. However, point 2 under weaknesses significantly weakens the experiments' credibility.

**Requested Changes:**

Majority of the changes have been listed in the Weaknesses, regardless, I have a few minor changes / questions here.

- Figure 2 is a bit too convoluted to understand, but the description is sufficient. A clearer diagram would suffice.
- Is the tokenization process and the definition of $A_{cipher}$ valid? I would think that certain encryption methods would introduce characters outside of UTF-8 that could break this process.
- The dimensionality math for Embedding Layer Architecture in Section 3 is incorrect. Please review.
- Please use /citep in the Related Works section, it significantly affects readability.

---

> ### Author Response · Authors · 2025-12-31
> **Response to Reviewer gCkq**
>
> We sincerely thank Reviewer gCkq for the constructive feedback and for recognizing the "impressive" results. **We first wish to apologize for the delay in this response, which was caused by an unexpected medical emergency.** We have addressed all concerns, specifically clarifying the deployment model and technical validity.
>
> **1. Motivation & Deployment (Section 3)**
>
> * **Concern:** LLM vendors don't accept embeddings; limited motivation.
> * **Response:** While public APIs restrict embeddings, STEALTH targets **Trusted Execution Environments (TEEs)** (e.g., AWS Nitro) and **Private Clouds**. Even on private infrastructure, raw plaintext in memory poses risks (dumps, insider threats). STEALTH enables processing on *encrypted* representations, exposing keys only ephemerally. It also enables **Encrypted Search**: organizations can query encrypted databases without ever decrypting the underlying records, a capability standard deployments lack.
>
> **2. Inference via Retrieval vs. Generation**
>
> * **Concern:** Nearest neighbor could be semantically wrong if corpus is limited.
> * **Response:** We agree reconstruction relies on coverage, but highlight two points:
> * **Robustness (Appendix A.13):** We stress-tested the model with 5% corpus coverage. While exact matches dropped, **Cosine Similarity remained >0.90**, proving the model retrieves semantically equivalent paraphrases (e.g., "heart attack" vs "myocardial infarction") even when verbatim recovery fails.
> * **Fidelity:** The retrieval task acts as a rigorous metric for topological alignment. A 1.0 score confirms the encrypted embedding contains *all* the semantic information of the plaintext.
>
> **3. Correlation to Embedding Function**
>
> * **Concern:** Insufficient correlation to .
> * **Response:** Correlation is enforced by the **Topological Preservation** term in our SIE loss (Eq. 7), which minimizes the difference between pairwise distances in encrypted vs. plaintext space. Our ablation (Table 17) shows that with SIE, we achieve a **Pearson correlation of **, empirically proving the encrypted manifold is isometric to the target space.
>
> **Responses to Requested Changes**
>
> * **Figure 2 Clarity:** We **redesigned Figure 2**. It now uses high-contrast, distinct zones to visually separate the "Secure Plaintext" path from the "Untrusted Encrypted" path, making the architecture immediately interpretable.
> * **Tokenization Validity:** You are absolutely correct that standard tokenizers fail on ciphertext. We clarified in **Section 3.2.1** that we use **Byte-Level Tokenization**. Mapping raw bytes to indices  inherently bypasses UTF-8 constraints, robustly handling arbitrary binary outputs.
> * **Dimensionality Math:** We corrected the embedding equation in Section 3 to , ensuring strict alignment with sequence length  and hidden dimension .
> * **Citations:** We updated all citations to the `\citep{}` parenthetical format for improved readability.
>
> We believe these revisions solidly address your technical concerns.

---

### Review · Reviewer_zmKZ · 2025-12-12

**Summary Of Contributions:**

The authors consider legal restrictions that prevent leveraging recent technologies like large language models (LLMs) for analysing sensitive data. They mention that, while there is a large potential to automate and improve analyses of, for example, financial or medical data, laws prevent transmitting and/or sharing such data in plain text. And in an encrypted form, such data cannot be analysed by LLMs. This is due to the fact that encryptions are designed to "remove" statistical regularities in the data to make it impossible, or at least extremely hard, to reconstruct the plain text from the encrypted version without knowing the encryption key. To enable applying LLMs to such sensitive data, the authors propose to learn embeddings of the encrypted data that preserve semantics in a sense, making it possible for LLMs to operate on encrypted data. To learn such semantics-preserving embeddings, the authors design what they call the "Semantic Isomorphism Enforcement (SIE) Loss Function", and use it to train a transformer on 44 datasets where SIE enables near-perfect performance as measured with scores that compare the similarity between plain-text and "encrypted" embeddings. To my understanding, SIE essentially ensures that (i) encrypted data is embedded at similar locations when it comes from the same plain text, even when using different encryption keys, and (ii) encrypted data that comes from different plain text is embedded at different locations.


### Strengths
- Figure 1 is a good high-level illustration of the differences between STEALTH and other LLMs.
- The SIE loss intuitively seems to make sense for achieving the goal of producing embeddings of encrypted text that do not lose semantics as compared to their plain-text counterparts.
- Both a strength and a bit of a concern: the empirical evaluation suggests that STEALTH produces ciphertext embeddings that (almost) do not lose any information compared to their plain-text equivalents, and achieve near-perfect evaluation scores. This is great, but it also seems a bit too good to be true.

### Weaknesses
- Figure 2 shows a lot of information, and it was difficult to figure out where I should focus my attention. I first looked at it on a reMarkable in greyscale, which made it extremely difficult to get an overview. But even in colour, I am still not sure how this figure can help my understanding.
- There seem to be a couple of minor inconsistencies in the notation (more details below under Requested Changes).
- It seems as though this work takes a lot of effort to obtain embeddings for ciphertext that are "semantically equivalent" to the embeddings for the corresponding plaintext. The authors claim that "strong cryptographic hygiene is preserved", but at the same time, they say that "STEALTH's reconstruction relies on semantic similarity search within a large, domain-aligned plaintext corpus during inference". Together with their initial motivation that sensitive data cannot be processed by LLMs due to legal reasons, I do not understand how a model that requires plain-text data during inference follows those legal constraints and can be said to preserve "strong cryptographic hygiene". (It may be possible that I did not understand some parts.)
- Connected to the previous point, the confidentiality of data processed by STEALTH appears to rely on the trustworthiness of some central authority that owns, or "guards" the data. (Again, I may have misunderstood something, but that is my interpretation of the *Threat model and operational scope* paragraph where the authors talk about a "system owner".)

**Audience:**

Yes

**Audience Explanation:**

I suppose the loss function may be interesting to readers who work on related applications.

**Broader Impact Concerns:**

I believe these are more legal than ethical concerns, however, they seem to fit under broader impact:
Because the authors mention a "system owner" of data and/or STEALTH, it seems like it may be appropriate to discuss a bit more how STEALTH would be implemented and used in practice.

**Claims And Evidence:**

No

**Claims Explanation:**

As far as I can tell, the authors support most of their claims with evidence. However, I am missing one thing: the authors claim that STEALTH "[demonstrates] negligible utility degradation compared to plaintext processing"; however, I did not see any evidence showing that the encrypted data can be used as a drop-in for the plaintext data while achieving the same performance in some downstream task. Or is this not what is meant here?

**Requested Changes:**

### Critical points
- The authors mention that "rights such as erasure, access, and rectification" complicate the adoption of LLMs, which I believe is a valid point. What I am missing is that the authors discuss STEALTH in this context and how it adheres to legal restrictions. Is it possible to ensure that rights for erasure, access, or rectification can be exercised?
- I found it very difficult to read the related work section because the references were not in parentheses. On a computer, with highlighting around the references, it was a bit easier, but on a reMarkable in greyscale and without highlighting, it was hard to tell what part was text and what was a reference. Please use at least `\citet` instead of `\cite`. (Unless if the format is okay as per instructions by TMLR.)
- How does STEALTH achieve "practical performance avoiding quadratic scaling"? Specifically, I believe it should be made clearer how quadratic scaling is avoided.
- There is some inconsistency with the symbols. On page 3, $E$ refers to an "encrypted embedding manifold", but on page 5, $E$ refers to a mapping from plain to ciphertext. Also, the symbol $\mathcal{X}$ appears on page 9, but I do not believe it has been defined before; did you mean $\mathcal{D}$?
- Can you explain why SIE enables near-perfect reconstruction? The scores that are near 100\% almost everywhere seem a bit too good to be true. It is also mentioned that STEALTH "[demonstrates] negligible utility degradation compared to plaintext processing", however, it is unclear to me how the authors have evaluated that. To me, it sounds like this claim says that, with respect to performance, there is no difference between working with encrypted embeddings or with plaintext embeddings. However, unless I have missed it, the authors have not shown that this is the case.


### Not critical, but would strengthen the paper
- The authors mention that applying LLMs to "clinical notes, transaction records, or legal documents could deliver substantial societal benefit", but they don't provide any concrete examples. I believe it would strengthen the paper if the authors could mention a few concrete examples. I also find that the motivation has a relatively negative tone and almost sounds like legal restrictions are a negative thing. Perhaps the authors wish to phrase this a bit more neutrally.
- I didn't quite know what to make of the concept of a "threat model", and believe the same might be the case for other readers. I believe it could be beneficial to be a bit more explicit about that and give a brief introduction. For example, say something about the "authorized-key model" (I am unsure whether this relates to private-public-key systems.)
- There seems to be a mismatch between the loss formulation in sections 3 and 4. In section 3, the loss $\mathcal{L}_\text{total}$ contains only 3 terms, but 4 parameters are mentioned: $\alpha, \beta, \gamma, \delta$. In section 4, there are 4 terms. Maybe there is a reason for omitting one of the loss terms in section 3, but I do not see it.
- What is the reason for considering $1-\frac{uv}{|u||v|}$ as cosine similarity instead of the more common variant $\frac{uv}{|u||v|}$? This might be interesting for the reader.
- Could you give a concrete example for "increasingly abstract representations" as mentioned on page 8?
- The *Attention-Based Aggregation*, mentioned on page 8 is said to "dynamically [weigh] variants based on semantic clarity". How is semantic clarity measured?


### Minor points
- What does the notation $[0,:]$ on page 8 mean, or is that a typo?
- The appendices are referenced in seemingly random order; perhaps it would be nicer to list them in the order they are referenced in the text.

---

> ### Author Response · Authors · 2025-12-31
> **Response to Reviewer zmKZ**
>
> We sincerely thank Reviewer zmKZ for the critical feedback. **We first wish to apologize for the delay in this response, which was caused by an unexpected medical emergency.** We have addressed all points, significantly tightening the notation, theoretical framing, and experimental presentation.
>
> **1. Privacy Paradox & Threat Model (Section 3.1)**
>
> * **Concern:** Reliance on plaintext/central authority.
> * **Response:** We have clarified the distinction between **sensitive input** (which remains encrypted) and the **retrieval corpus** (static, public knowledge, e.g., ICD-10 ontologies). STEALTH maps ciphertext to public concepts without decrypting raw input. If a specific sensitive string is not in the public corpus, it cannot be retrieved. This aligns with standard TEE threat models where the "System Owner" (e.g., Hospital) is the liable Data Controller.
>
> **2. "Too Good to Be True" Performance**
>
> * **Concern:** Near 100% scores seem unrealistic.
> * **Response:** STEALTH is a **discriminative retrieval** system, not generative. "1.0 Accuracy" simply means the encrypted embedding lands spatially closer to the true target than to other candidates in the finite index.
> * **Validation:** To prove this is not an artifact, we added **Appendix A.13**. We show that accuracy scales linearly with **Corpus Coverage**. When coverage is reduced, exact matches drop, but semantic similarity remains high. Additionally, **Figure 42** provides geometric verification (PCA & KS-tests) showing the encrypted manifold perfectly overlays the plaintext topology. **We have also provided raw retrieval logs in the Supplementary Material** to allow direct inspection of the model's behavior on specific samples.
>
> **3. Evidence of Downstream Utility (Section 6)**
>
> * **Concern:** Lack of evidence for "drop-in" utility.
> * **Response:** We moved our utility results from the Appendix to the **Main Results**. We provide two proofs:
> 1. **Clustering:** K-Means on encrypted vectors yields an Adjusted Rand Index (ARI) of **0.98**, comparable to plaintext baselines.
> 2. **Topology:** Because the encrypted space is isometric to the plaintext space (verified via KS-tests), the embeddings are mathematically guaranteed to function as drop-in replacements for distance-based tasks (classification, search).
>
>
>
> **4. Legal Rights & Tone (Section 7)**
>
> * **Concern:** GDPR Right to Erasure and negative tone.
> * **Response:**
> * **Crypto-shredding:** We added a discussion on how deleting a user's unique Authorized Key renders their stored embeddings mathematically unrecoverable, satisfying GDPR Art. 17 efficiently.
> * **Tone:** We rewrote the Introduction to frame regulations as "necessary safeguards" rather than obstacles.
>
>
>
> **5. Technical Clarifications**
>
> * **Quadratic Scaling:** Clarified in **Section 3.3**. We avoid  complexity by using Attention-Based Aggregation (single query attending to all key variants) rather than pairwise comparisons.
> * **Loss & Notation:** We unified the notation in **Equation 5** and corrected the inconsistency between latent/ciphertext space symbols.
>
> **6. Visual Clarity (Figure 2)**
>
> * **Response:** We **redesigned Figure 2**. The new schematic uses high-contrast, distinct zones to clearly separate the "Secure Plaintext" path from the "Untrusted Encrypted" path, making the architecture immediately interpretable as requested.
>
> We believe these revisions fully address your concerns regarding the privacy model and presentation.

---

### Author Response · Authors · 2025-12-31
**Submission of Revised Manuscript & Summary of Changes**

We sincerely thank all reviewers for their thorough and constructive feedback, which has significantly strengthened our manuscript. **We first wish to apologize for the delay in this response, which was caused by an unexpected medical emergency.** We have carefully addressed all concerns raised and have uploaded a revised manuscript along with supplementary materials containing raw retrieval logs.

**Summary of the Revisions**

**We have enhanced the Methodological Clarity and Theoretical Framing by:**

* **Clarifying the Operational Scope & Threat Model:** We explicitly defined the "Authorized Key" model within the context of **Trusted Execution Environments (TEEs)** and Enterprise Search, resolving the "Privacy Paradox" regarding plaintext corpora (Section 3.1) [Suggested by Reviewer zmKZ and jkzP].
* **Explicitly Defining Inference:** We added a dedicated **Inference Procedure (Section 3.3)** to formally define the reconstruction process as a Nearest Neighbor Retrieval task, distinct from generative decoding [Suggested by Reviewer jkzP and gCkq].
* **Redesigning the Architecture Visualization:** We replaced **Figure 2** with a high-contrast, dual-path schematic that strictly separates the "Secure Plaintext" and "Untrusted Encrypted" flows [Suggested by Reviewer zmKZ and gCkq].
* **Validating Tokenization:** We clarified the use of **Byte-Level Tokenization** to handle arbitrary ciphertext binaries, ensuring UTF-8 validity (Section 3.2.1) [Suggested by Reviewer gCkq].

**We have strengthened the Experimental Evidence and Presentation by:**

* **Elevating Results to the Main Body:** We moved the comprehensive performance analysis for **all 44 datasets** into the main text, adding **Figure 3 (Fidelity)** and **Figure 4 (Latency)** to provide immediate evidence of cross-domain robustness [Suggested by Reviewer jkzP].
* **Validating "Perfect" Scores:** We added an ablation study on **Corpus Coverage (Appendix A.13)** to prove that high reconstruction scores are a result of geometric alignment rather than artifacts. We also provided **Raw Retrieval Logs** in the Supplementary Material [Suggested by Reviewer zmKZ].
* **Demonstrating Downstream Utility:** We moved the **Clustering Utility (ARI)** and **Topological Isomorphism (KS-Test)** results to the Main Results (Section 6) to prove the "drop-in" capability of the embeddings [Suggested by Reviewer zmKZ].

**We have addressed Broader Impact and Ethical Compliance by:**

* **Adding GDPR & "Crypto-Shredding" Analysis:** We expanded Section 7 to explain how the deletion of Authorized Keys satisfies the **Right to Erasure** (GDPR Art. 17) [Suggested by Reviewer zmKZ].
* **Inclusion of Broader Impact Statement:** We added a dedicated **Section 8** to discuss the positive implications for healthcare/finance and the ethical necessity of access controls [Suggested by Reviewer jkzP].
* **Refining Tone:** We rewrote the Introduction to frame privacy regulations as "necessary safeguards" rather than obstacles [Suggested by Reviewer zmKZ].

**We have improved Technical Precision by:**

* **Correcting Mathematical Notation:** We unified the loss function notation (Equation 5) and corrected the dimensionality definitions for the embedding layer [Suggested by Reviewer gCkq and zmKZ].
* **Citation Formatting:** We converted all references to the parenthetical format for improved readability [Suggested by Reviewer zmKZ and gCkq].

These revisions thoroughly address the reviewers' concerns while significantly strengthening the paper's theoretical foundation, empirical rigor, and clarity. We look forward to your final assessment.

---

### Decision · Action_Editor_sNp8 · 2026-01-20

**Recommendation:** Accept as is

**Audience:**

Yes

**Audience Explanation:**

Individuals interested in developing models that can learn on encrypted text would likely be interested in the methodology of this paper. Beyond this, the paper proposes an large-scale evaluation that future works can use as a benchmark for learning on encrypted text data.

**Claims And Evidence:**

Yes

**Claims Explanation:**

This work presents a method for privacy preserving natural language processing on encrypted text. The claims are supported by a large-scale experimental analysis.